# Prevalence and determinants of poor glycemic control among diabetic chronic kidney disease patients on maintenance hemodialysis in Tanzania

Emmanuel Arthur Mfundo[1,2], Alphonce Ignace Marealle[2], Goodluck G. Nyondo[3], Martine A. Manguzu[2], Deus Buma[4], Peter Kunambi[5], Ritah F. Mutagonda[2]*

1 Quality Assurance Department, National Health Insurance Fund, Arusha, Tanzania, 2 Department of Clinical Pharmacy and Pharmacology, Muhimbili University of Health and Allied Sciences, Dar es Salaam, Tanzania, 3 Department of Medicinal Chemistry, School of Pharmacy, Muhimbili University of Health and Allied Sciences, Dar es Salaam, Tanzania, 4 Department of Clinical Research, Training and Consultancy, Muhimbili National Hospital, Dar es Salaam, Tanzania, 5 Department of Clinical Pharmacology, Muhimbili University of Health and Allied Sciences, Dar es Salaam, Tanzania

* ritah.mutagonda@muhas.ac.tz

## Abstract

### Background

Poor glycemic control in diabetic chronic kidney disease (CKD) patients on maintenance hemodialysis is of great challenge, resulting in increased risk of morbidity and mortality. This study aimed to determine the prevalence and determinants of poor glycemic control among diabetic CKD patients on maintenance hemodialysis.

### Methodology

A cross-sectional study was conducted in 12 dialysis centers located in four regions of Tanzania from March to June 2023. The study population was diabetic CKD patients above 18 years on maintenance hemodialysis for three months or more. A consecutive sampling technique was used for patient recruitment, and a semi-structured questionnaire was used to collect data. The primary outcome was poor glycemic control were considered when glycated hemoglobin (HbA1c) levels were < 6% or >8%. Statistical Package for Social Sciences (SPSS) version 23 was used for data analysis. Univariate and multivariable regression models were used to evaluate the determinants of poor glycemic control. A p-value <0.05 was considered statistically significant.

### Results

Out of 233 enrolled patients, the overall prevalence of poor glycemic control was 55.4%, whereby 27.0% had HbA1c <6% and 28.33% had HbA1c >8%. A high risk of HbA1c >8% was observed among patients who were on antidiabetic medication (2.16 (95% CI: 1.06–4.41) p = 0.035) and those attending dialysis sessions less than 3 times a week (1.59 (95% CI: 1.02–2.48) p = 0.040). The lower risk of HbA1c <6% was observed in patients

**Data availability statement:** Dataset has been attached as Supporting Information.

**Funding:** The author(s) received no specific funding for this work.

**Competing interests:** The authors declare that there is no competing interest.

dialyzed using glucose-containing dialysates than those dialyzed with glucose-free dialysate (0.57 (95% CI 0.36–0.87) p = 0.020).

## Conclusion

The high prevalence of poor glycemic control among diabetic CKD patients, as revealed by this study, has significant implications. Patients on antidiabetic medication and those with less than three dialysis sessions per week are at a high risk of HbA1c >8%. Conversely, patients dialyzed using glucose-free dialysates are at a high risk of HbA1c <6%. Glycemic control in diabetic chronic kidney disease (CKD) patients is a great challenge due to altered glucose homeostasis, gluconeogenesis, tubular glucose reabsorption and inaccuracy of glycemic regulation metrics [1]. Furthermore, changed renal pharmacokinetics of antihyperglycemic agents (AHA), uremic milieu, and dialysis therapy also contribute to this challenge [2]. Based on the severe risk of hyperglycemia and hypoglycemia in patients with diabetic end-stage renal disease (ESRD), glycemic control is of paramount importance.

## Background

A study conducted in India showed a higher frequency of hypoglycemia in diabetic CKD patients compared to those without CKD. Moreover, a high mortality rate was associated with hypoglycemia among these patients [3]. In addition, a significant association has been reported between poor glycemic control and low survival rates in diabetic CKD patients on hemodialysis compared to individuals with proper glycemic control. Notably, cardiovascular and infections were significant causes of death in this patient population [4]. Furthermore, alteration in metabolism and elimination of antihyperglycemic agents (AHA) contribute to the incidence of hypoglycemia in diabetic hemodialysis patients [3,5].

A study conducted in Canada reported a 54% prevalence of poor glycemic control among diabetic patients undergoing hemodialysis [6]. In addition, another study conducted in Tanzania reported glycemic control to be positively linked with adherence to antidiabetic medications, where optimum glycemic control was achieved in patients with good adherence to medication [7]. In contrast, poor glycemic control is associated with non-adherence to antidiabetic medications, leading to hyperglycemia-related complications such as coronary artery disease (CAD), cardiovascular disease (CVD), hypertension, dyslipidemia, poor patient quality of life, and prolonged hospital admissions, which increase healthcare costs [8].

The Tanzanian guideline on hemodialysis recommends all patients have baseline measurements of their blood sugar level and glycated hemoglobin (HbA1c) at the time of initiation of dialysis therapy and recommends HbA1c between 7% and 8% as an ideal target. Furthermore, KDIGO clinical practice guidelines for diabetic management in CKD (2022) recommend an ideal HbA1c between 6% and 8% for diabetic CKD patients on maintenance hemodialysis [9]. This is also supported by the previous epidemiological study, which reported that HbA1c from 6% to 8% is associated with decreased mortality rates [2].

Several factors determine the glycemic control of diabetic patients with or without renal failure, which includes eating a healthy diet, physical exercise, blood transfusion, elevated blood urea nitrogen (BUN), hemodialysis procedures, use of EPO, medication dose, and adherence [2,10]. Accumulation of uremic toxins among diabetic CKD patients compromises the quality of life due to increased risk of uremia, development of insulin resistance, decreased hepatic insulin metabolism, uremic malnutrition, anemia, gastritis, and increased risk of

hospitalization due to cognitive impairment [11,12]. Therefore, adequately removing uremic toxins and other waste products from the patient's blood through hemodialysis improves the quality of life and survival rate of this patient group.

Despite the importance of glycemic control to diabetic CKD patients on maintenance dialysis, limited data exist in sub-Saharan Africa, including Tanzania. Therefore, this study aimed to evaluate the prevalence and determinants of poor glycemic control among diabetic CKD patients attending hemodialysis centers in Tanzania.

## Methods

### Study design and settings

A cross-sectional study was conducted at selected hemodialysis centers in Tanzania from 1st March to 30th June 2023. The enrolling sites were Muhimbili National Hospital (MNH) Upanga and Mloganzila centers, CCBRT Hospital, Hubert Kairuki Hospital, TMJ Hospital, Hindu Mandal Hospital and Hindu Mandal Polyclinic, in Dar es Salaam region, Benjamin Mkapa Hospital found in Dodoma, NSK Hospital, Mt. Meru Hospital and Moyo Medicare specialized polyclinic in Arusha region and Bugando Medical Centre in Mwanza region. These centers were selected because they serve most patients from all parts of Tanzania, and they were confirmed to monitor patients' glycemic control using glycated hemoglobin assay. In addition, in all these centers, diabetic CKD patients are under the care of the nephrologists.

### Study population

The study population consisted of diabetic CKD patients on maintenance hemodialysis at dialysis centers in Tanzania.

### Inclusion and exclusion criteria

The study included diabetic hemodialysis patients aged ≥ 18 years on maintenance hemodialysis for ≥ 3 months. It excluded critically ill patients who failed to express and consent themselves, those who voluntarily refused to be included, and those who had not performed an HbA1c test within three months. Excluding patients without recent HbA1c results was essential to ensure data accuracy, consistency, and reliability while reducing the risk of biases such as misclassification, selection, and confounding.

### Sample size calculation

A sample size of 380 participants was obtained using a prevalence of 54% poor glycemic control in diabetic patients on hemodialysis conducted in Canada with a 95% confidence level and a margin of error of 5% [12]. However, due to a restricted number of diabetic hemodialysis patients at dialysis centers in Tanzania, the sample size was recalculated using a finite population obtained from the diabetic-CKD population at thirteen [12] data collection sites, which was 600, yielding a sample size of 238 people. The recalculation yielded a sample size of 238 people, whereby 5 patients were excluded from the study due to abnormal records of HbA1c due to incorrect documentation. Therefore, only 233 patients were included in the final analysis.

### Sampling strategy

Purposive sampling of regions and health facilities was used to enroll facilities providing dialysis services and having the laboratory capacity to monitor glycemic control regularly. Then a consecutive sampling technique was used to enroll patients in the selected facilities.

## Data collection

A semi-structured questionnaire was used to collect primary data from patients, including sociodemographic factors, diabetes duration and dialysis treatment, weight, height, and diabetes-related comorbidities. Secondary data from the patient's hospital records included the patient's medical history and type of investigations, hemoglobin levels (three months), glycated hemoglobin (one month), blood urea nitrogen (BUN) (three months), serum creatinine (one month), diabetic medications, and other concomitant medications. The tool was reviewed by experts on the subject to ensure relevance, clarity and comprehensiveness of the questions was assessed. Then the tool was pre-tested to a small group of diabetic CKD patients on hemodialysis whereby the results of the pretest were further used to refine the tool.

## Data analysis

Data were analyzed using Statistical Package for Social Sciences (IBM-SPSS-version 23). Continuous variables were summarized using median and categorical variables were summarized using frequency and percentages. Variables were selected based on their potential association with glycemic control. Hemoglobin (Hb) levels were graded as either anemic or non-anemic based on Tanzania national dialysis guideline (Low Hb < 10g/dl, Normal > 10g/dl). Average pre- and post-blood urea nitrogen (BUN) levels were used to compute urea reduction ratio (URR) as a maker for dialysis adequacy and a dialysis adequacy was regarded as urea reduction ratio (URR) of > 65%. Furthermore, the Cockcroft and Gault equation was used to calculate estimated glomerular filtration (eGFR), and eGFR of < 15ml/minutes was regarded as kidney failure stage five [13].

Based on KDIGO clinical practice guidelines for diabetic management in CKD [9], HbA1c between 6% and 8% was considered adequate, whereas HbA1c levels < 6% and > 8% were regarded as poor glycemic control. Prevalence ratios and their corresponding 95% confidence intervals (CI) were calculated by using univariate and multivariable regression models to show the determinants of poor glycemic control. In a multivariable analysis, potential confounders were controlled by including them as covariates (independent variables) in the regression model. This statistical analysis isolated the effect of the independent variables such as age, dialysis duration, comorbidities etc. on the outcome of interest (glycemic control) while accounting for other factors that might influence the outcome. Variables with $p < 0.1$ on univariate analysis were considered for multivariate regression. A p-value of $< 0.05$ was considered statistically significant.

## Ethics approval and consent to participate

This study obtained ethical clearance from Muhimbili University of Health and Allied Sciences (MUHAS) institutional review board (Certificate number: MUHAS-REC-02-2023-1533). In addition, permission to conduct this study in the selected facilities was sought from each hospital management. Written informed consent was obtained from each participant.

# Results

## Baseline socio-demographic characteristics of the study participants

Among the study participants (n = 233), 70.4% were male, and the majority, 60.1%, were aged > 60 years with a median age of 63 years, IQR (56, 68). About 46% of participants were retirees, and 96.6% had insurance, Table 1.

## Clinical characteristics of the study participants

The study participants were comprised of 96.1% with type 2 diabetes, 56.2% with a family history of diabetes, 99.1% hypertensive, and 26.6% stopped antihyperglycemic agents due to

Table 1. Baseline characteristics of the study participants (N = 233).

| Variable | Frequency (n) | Percent (%) |
|---|---|---|
| **Age group (years)** | | |
| 18–60 | 93 | 39.9 |
| >60 | 140 | 60.1 |
| Median age in years (IQR) | 63 (56, 68) | |
| **Sex** | | |
| Male | 164 | 70.4 |
| Female | 69 | 29.6 |
| **Marital status** | | |
| Married | 203 | 87.1 |
| Not married | 30 | 12.9 |
| **Education Level** | | |
| No formal to primary education | 76 | 32.6 |
| Secondary education | 42 | 18.0 |
| Vocational training/certificate | 24 | 10.3 |
| Diploma | 38 | 16.3 |
| Degree/Masters/PhD | 53 | 22.7 |
| **Occupation** | | |
| Not employed | 41 | 17.6 |
| Formal employment | 43 | 18.5 |
| Business | 39 | 16.7 |
| Student | 2 | 0.9 |
| Retired | 108 | 46.4 |
| **History of cigarette smoking** | | |
| Ever smoked | 60 | 25.8 |
| Never smoked | 173 | 74.2 |
| **History of alcohol use** | | |
| Ever drunk | 154 | 66.1 |
| Never drunk | 79 | 33.9 |
| **Payment mode** | | |
| Insurance | 225 | 96.6 |
| Cash | 8 | 3.4 |

the resolution of their hyperglycemia for more than one year. The majority (93.6%) of participants had CKD stage five, 88.4% were on dialysis three times per week, and 77.2% (n = 202) had attained dialysis adequacy. In addition, 89.3% reported measuring blood glucose at home as a continuous self-monitoring of diabetes, and 76% had developed diabetes-related complications such as peripheral neuropathy (72.1%), retinopathy (34.3%), micro/macrovascular (9.4%), and gastroparesis (3.4%). About 47.2% of patients had an arterio-venous fistula (AV-fistula) as vascular access. Table 2.

## Prevalence of poor glycemic control

This study results revealed a prevalence of 55.4% of poor glycemic control among diabetic CKD patients on maintenance dialysis, which is constituted by 27.04% of participants with HbA1c < 6% and 28.33% with HbA1c > 8%).

**Table 2. Clinical characteristics of the study participants (N = 233).**

| Variable | Frequency (n) | Percent (%) |
|---|---|---|
| **Type of Diabetes** | | |
| Type 1 | 9 | 3.9 |
| Type 2 | 224 | 96.1 |
| **Family history of diabetes** | | |
| Yes | 131 | 56.2 |
| No | 102 | 43.8 |
| **History of hypertension** | | |
| Yes | 231 | 99.1 |
| No | 2 | 0.9 |
| **Testing blood sugar at home** | | |
| Yes | 208 | 89.3 |
| No | 25 | 10.7 |
| **Blood sugar readings (n = 208)** | | |
| Usually high | 9 | 4.3 |
| Usually low | 1 | 0.5 |
| Within goals | 97 | 46.6 |
| Fluctuating between high and low | 101 | 48.6 |
| **Experience episode of hypoglycemia** | | |
| Yes | 125 | 53.6 |
| No | 108 | 46.4 |
| **BMI (kg/m²)** | | |
| Underweight (< 18.5) | 13 | 5.6 |
| Normal weight (18.5–24.9) | 118 | 50.6 |
| Overweight (25.0–29.9) | 75 | 32.2 |
| Obesity (≥ 30.0) | 27 | 11.6 |
| **When does hypoglycemia occur (n = 125)** | | |
| During dialysis | 2 | 1.6 |
| After dialysis | 4 | 3.2 |
| At Home | 119 | 95.2 |
| **Diabetic induced complications** | | |
| Yes | 177 | 76.0 |
| No | 56 | 24.0 |

## Determinants of glycemic state in diabetic hemodialysis patients

Patients who were still on antidiabetic medications had higher risk of poor glycemic control (HbA1c > 8%) compared to those who stopped using medication due to resolution of hyperglycemia 2.16 (95% CI 1.06–4.41) p = 0.035. Patients attending dialysis sessions once to twice a week had a higher risk of poor glycemic control (HbA1c > 8%) than those attending thrice a week 1.59 (95 CI 1.02–2.48), p = 0.040 (Table 3).

Patients dialyzed using glucose-containing dialysates had a lower risk of poor glycemic control (HbA1c < 6%) than those dialyzed with glucose-free dialysate 0.57 (95 CI 0.35–0.92), p = 0.020, and Table 4.

## Discussion

Our study aimed to determine the prevalence and determinants of poor glycemic control among diabetic CKD patients on maintenance hemodialysis. Significant findings of this study

**Table 3. Univariable and multivariable analysis for predictors associated with high HbA1c >8% levels among diabetic_CKD patients on hemodialysis (n = 170).**

| Variable | Category | Univariable analysis | | Multivariable analysis | |
|---|---|---|---|---|---|
| | | cPR (95% CI) | P - value | aPR (95% CI) | P – value |
| **Mode of payment** | Insurance | 0.6 (0.34–1.07) | **0.083** | 0.93 (0.46–1.88) | 0.847 |
| | Cash | 1ᵃ | | | |
| **DM medication status** | On medication | 2.23 (1.06–4.69) | **0.034** | 2.16 (1.06–4.41) | **0.035** |
| | Not on medication | 1ᵃ | | | |
| **Insulin use** | Yes | 1.27 (0.86–1.87) | **0.226** | 0.99 (0.66–1.49) | 0.968 |
| | No | 1ᵃ | | | |
| **Blood sugar reading at home** | Within goals | 0.56 (0.35–0.90 | **0.017** * | | |
| | Fluctuating | 1ᵃ | | | |
| **Episodes of hypoglycemia** | Yes | 1.37 (0.92–2.06) | 0.124 | 1.20 (0.81–1.79) | 0.362 |
| | No | | | | |
| **URR (%)** | Inadequacy < 65 | 1.34 (0.86–2.07) | 0.192 * | | |
| | Adequacy ≥65 | 1ᵃ | | | |
| **Dialysis frequency (per week)** | Once to twice | 1.64 (1.09–2.45) | **0.016** | 1.59 (1.02–2.48) | **0.040** |
| | Thrice | 1ᵃ | | | |
| **Type of dialysate** | With glucose | 0.61 (0.39–0.97) | **0.038** | 0.71 (0.45–1.11) | 0.136 |
| | Glucose free | 1ᵃ | | | |
| **BT for past 3months** | Yes | 1.76 (0.97–3.21) | 0.063 | 1.55 (0.87–2.78) | 0.139 |
| | No | 1ᵃ | | | |
| **Vitamin D use** | Yes | 0.67 (0.46–0.97) | **0.033** | 0.72 (0.49–1.06) | 0.094 |
| | No | 1ᵃ | | | |
| **Suffered infection** | Yes | 1.51 (0.95–2.39) | 0.082 | 1.46 (0.92–2.33) | 0.109 |
| | No | 1ᵃ | | | |

include a high prevalence (55.4%) of poor glycemic control among diabetic CKD patients in the population. It was observed that being on medication and attending dialysis sessions only once to twice a week were significantly associated with HbA1c >8% levels, while those who were dialyzed using glucose-containing dialysates were significantly associated with HbA1c <6% levels.

The high prevalence of poor glycemic control in this study is in line with the findings from a study conducted in Canada, which reported a prevalence of 54% among patients who developed diabetic-related complications [14]. Inadequate glycemic control with HbA1c >8% can contribute to the development of diabetic-related complications such as diabetic neuropathy, retinopathy, and macro and microvascular complications. This was also evident in our study, whereby 76% of patients had diabetic-induced complications such as diabetic neuropathy, retinopathy and macro/microvascular complications [14]. The HbA1c <6% level is known to contribute to increased mortality among diabetic CKD patients on maintenance hemodialysis. Two studies conducted in London, United Kingdom, investigated the association between HbA1c and mortality rate and showed a J-shaped distribution of mortality risks. The studies concluded that both deficient and high levels of HbA1c were related to increased mortality risks due to diabetic complications and hypoglycemia, respectively [15,16]. Further studies need to be conducted in resource-constrained countries to investigate the risk of mortality in this group of patients.

This study showed that patients who were still on antidiabetic medication had a significantly higher risk of having HbA1c >8% levels compared to those who had their medications stopped due to the resolution of hyperglycemia. This might be due to insulin resistance associated with the

**Table 4. Univariable and multivariable analysis for predictors associated with low HbA1c < 6% among diabetic-CKD patients on hemodialysis (n = 167).**

| Variable | Category | Univariable analysis | | Multivariable analysis | |
|---|---|---|---|---|---|
| | | cPR (95% CI) | P – value | aPR (95% CI) | P – value |
| **Marital status** | Not married | 0.47 (0.19–1.16) | 0.103 | 0.61 (0.27–1.36) | 0.228 |
| | Married | 1ᵃ | | | |
| **DM medication status** | On medication | 0.52 (0.36–0.76) | **0.001** | 0.69 (0.37–1.29) | 0.245 |
| | Not on medication | 1ᵃ | | | |
| **Insulin use** | Yes | 0.69 (0.39–1.21) | 0.193 | 0.78 (0.39–1.53) | 0.472 |
| | No | 1ᵃ | | | |
| **Episodes of hypoglycemia** | Yes | 0.78 (0.52–1.16) | 0.213 | 0.88 (0.58–1.33) | 0.547 |
| | No | | | | |
| **Type of dialysate** | With glucose | 0.56 (0.36–0.87) | **0.010** | 0.57 (0.35–0.92) | **0.020** |
| | Glucose free | 1ᵃ | | | |
| **Sulfonylurea use** | Yes | 0.61 (0.38–0.97) | 0.038 | 0.79 (0.42–1.49) | 0.473 |
| | No | 1ᵃ | | | |
| **BT for past 3months** | Yes | 1.62 (0.77–3.4 1) | 0.203 | 1.99 (0.98–4.05) | 0.059 |
| | No | 1ᵃ | | | |
| **Duration on dialysis (Months)** | < 6.0 | 0.90 (0.32–2.54) | **0.840** | 1.04 (0.32–3.35) | 0.953 |
| | 6.0–11.0 | 0.67 (0.39–1.56) | **0.149** | 074 (0.44–1.24) | 0.254 |
| | 12.0–35.0 | 0.54 (0.32–0.91) | **0.019** | 0.61 (0.36–1.01) | 0.056 |
| | 36.0–59.0 | 0.66 (0.39–1.09) | **0.108** | 0.65 (0.40–1.07) | 0.088 |
| | ≥ 60.0 | 1ᵃ | | | |
| **Calcium channel blocker use** | Yes | 0.72 (0.45–1.15) | 0.167 | 0.62 (0.36–1.05) | 0.074 |

cumulation of uremic toxins in this patient's group and non-adherence to medication. Our study indicated that 33.9% of patients had non-adherence to medication, with about 55.2% of these patients having poor glycemic control. Our study findings correlate with the study conducted in Canada in which more patients who were on medication had poor glycemic control [14].

For the case of HbA1c < 6%, our study showed that patients who were dialyzed using glucose-containing dialysate had a lower risk of developing inadequate low glycemic control compared to those patients dialyzed without glucose-containing dialysate. This is because some of the blood glucose is lost during dialysis procedures due to hemofiltration [17]. Our study findings correlate with the study conducted in India, which reported higher hypo-glycemic episodes in patients dialyzed with glucose-free dialysates when compared with glucose-containing dialysate [18]. Despite some potential benefits of using glucose-free dialysates, such as reducing risks of bacterial infection, reducing skin autofluorescence (SAF), which is a marker of advanced glycation end products (AGEs), mobilizing lipid to maintain normal blood glucose [19], in order to avoid the risk of hypoglycemic episodes and extremely low glucose levels, glucose-containing dialysates are recommended.

Moreover, the study showed that patients having once to twice hemodialysis sessions per week had a higher risk of HbA1c > 8% compared to those with three sessions per week. Dialysis frequency may be influenced by several factors but mainly the ability to pay for dialysis sessions. Though during the study period, 96.6% of patients were covered with insurance, previous studies conducted in Tanzania have reported that the frequency of maintenance hemodialysis is significantly influenced by the mode of payment, whereby patients covered by health insurance had higher adherence to dialysis sessions compared to non-insured patients [20,21]. More studies can be conducted to provide evidence on factors influencing dialysis frequency among insured patients.

## Study limitations

There were certain drawbacks in this study, such as participants' reluctance to provide the essential information and economic restraints, which caused some of them to fail to pay for the prescribed three weekly hemodialysis sessions and HbA1c assays. The study excluded patients who did not have recent HbA1c (collected within three months before the study). Excluding such patients could have resulted in selection bias, whereby patients without HbA1c results may systematically differ from those with recent results. However, for the purpose of this study including them without valid HbA1c data could have diluted or confounded the true relationships between the studied variables. Therefore, we recommend further studies to explore more of the factors that could affect adequate monitoring of glycemic control among this study population.

## Conclusions

This study indicates that the prevalence of poor glycemic control in Tanzania is high among diabetic CKD patients on maintenance hemodialysis. Patients who had less than three dialysis sessions per week had HbA1c > 8%. We recommend tailored interventions to ensure that diabetic CKD receive three dialysis sessions per week. Moreover, patients who were dialyzed using glucose-free dialysates had poor low glycemic control, which emphasizes the significance of using glucose-containing dialysates in our settings.

## Supporting information

**S1 File. Data set.**
(XLSX)

## Acknowledgments

The authors acknowledge the management of National Health Insurance Fund (NHIF), staff and management of the data collection sites, Mr. Victor Njau, Mr. Devis Mhagama and staff of the Department of Clinical Pharmacy and Pharmacology, MUHAS.

## Author contributions

**Conceptualization:** Emmanuel Arthur Mfundo, Alphonce Ignace Marealle, Deus Buma, Ritah F. Mutagonda.

**Data curation:** Emmanuel Arthur Mfundo, Alphonce Ignace Marealle, Goodluck G. Nyondo, Martine A. Manguzu, Peter Kunambi, Ritah F. Mutagonda.

**Formal analysis:** Emmanuel Arthur Mfundo, Peter Kunambi, Ritah F. Mutagonda.

**Investigation:** Emmanuel Arthur Mfundo.

**Methodology:** Emmanuel Arthur Mfundo, Alphonce Ignace Marealle, Ritah F. Mutagonda.

**Project administration:** Emmanuel Arthur Mfundo.

**Resources:** Deus Buma.

**Supervision:** Alphonce Ignace Marealle, Deus Buma, Ritah F. Mutagonda.

**Validation:** Alphonce Ignace Marealle, Goodluck G. Nyondo, Martine A. Manguzu, Deus Buma, Ritah F. Mutagonda.

**Visualization:** Alphonce Ignace Marealle, Goodluck G. Nyondo, Martine A. Manguzu, Deus Buma, Ritah F.. Mutagonda.

**Writing – original draft:** Emmanuel Arthur Mfundo.

**Writing – review & editing:** Emmanuel Arthur Mfundo, Alphonce Ignace Marealle, Goodluck G. Nyondo, Martine A. Manguzu, Deus Buma, Peter Kunambi, Ritah F. Mutagonda.

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
