## [Decision Letter · Decision Letter 0]

30 Jul 2024

PONE-D-24-23929Prevalence and determinants of poor glycemic control among diabetic chronic kidney disease patients on maintenance hemodialysis in Tanzania

PLOS ONE

Dear Dr. Mutagonda,

Thank you for submitting your manuscript to PLOS ONE. After careful consideration, we feel that it has merit but does not fully meet PLOS ONE’s publication criteria as it currently stands. Therefore, we invite you to submit a revised version of the manuscript that addresses the points raised during the review process.

Sample size calculation, In this section you mentioned the sample size is 380 but in the result you said n=233.

In the table 3 you mentioned the sample is 170.

Please mention the chart flow of this study in the method section including the sample that you have used

We look forward to receiving your revised manuscript.

Kind regards,

Diana Laila Ramatillah, PhD

Academic Editor

PLOS ONE

Additional Editor Comments:

Sample size calculation

In this section you mentioned the sample size is 380 but in the result you said n=233.

In the table 3 you mentioned the sample is 170.

Please mention the chart flow of this study in the method section including the sample that you have used

Reviewers' comments:

Reviewer's Responses to Questions

**Comments to the Author**

1. Is the manuscript technically sound, and do the data support the conclusions?

Reviewer #1: Partly

Reviewer #2: Yes

2. Has the statistical analysis been performed appropriately and rigorously?

Reviewer #1: Yes

Reviewer #2: I Don't Know

3. Have the authors made all data underlying the findings in their manuscript fully available?

Reviewer #1: Yes

Reviewer #2: Yes

4. Is the manuscript presented in an intelligible fashion and written in standard English?

Reviewer #1: Yes

Reviewer #2: Yes

5. Review Comments to the Author

Reviewer #1: This manuscript describes the Prevalence and determinants of poor glycemic control among diabetic chronic kidney disease patients on maintenance hemodialysis in Tanzania.

This article is fairly well written but could use one more review by a native English speaker as there is unnecessary addition of commas in several areas and word "the" appears to be missing in several areas. In addition, some of the sentence are a little confusing and seem like the words were mixed up.

Major comment:

The range of HBA1c set by the author is unclear to me. There are a lot of papers that consider HBA1c of <6 % to be normal. The Lancet laboratories Tanzania consider the HBA1C less than 5.7% to be normal. HBA1c level of less than 5% is considered risk for diabetes patients.

https://www.cerbalancetafrica.co.tz/media/5kih0nky/diabetes-panel-tanzania.pdf

Overall, these is a very good start here. In order for this to be a valuable contribution to published literature, I would encourage the authors to take a few more steps to increase the impact of their research.

Abstract:

Minimize the use of commas in the abstract and make sentences easier to grasp.

In Results

Please rewrite the line “While the predictor of HbA1c <6% was the type of dialyzer used (0.57 (95% CI 0.36 – 0.87) p = 0.020)”. And information regarding HBA1c <6 is not sufficient.

Background

Line 22- Hyper dislipidemia should be written either dyslipidemia or hyperlipidemia.

Line 26-28- Please reconsider writing these lines as these lines are difficult to understand. Please verify either these lines are according to WHO guidelines for Diabetes Mellitus or not.

In Results:

Line 104- 88.4 should be written as 88.4%

Best regards,

Reviewer

Reviewer #2: My general comment:

The manuscript titled "Prevalence and determinants of poor glycemic control among diabetic chronic kidney disease patients on maintenance hemodialysis in Tanzania" is a well-structured and thorough study addressing an important public health issue. The authors have effectively described their methodology, provided clear results, and discussed significant findings.

Here are some specific comments and suggestions for improvement:

Title

Title Appropriateness: The title is clear and descriptive, accurately reflecting the content and focus of the study.

Abstract

Clarity and Conciseness: The abstract effectively summarizes the study's background, methodology, results, and conclusions. It is concise and provides a good overview of the research.

* Minor Typo: Change "was considered when glycated hemoglobin (HbA1c) levels were < 6% or >8%" to "were considered when glycated hemoglobin (HbA1c) levels were < 6% or >8%."

Introduction

Context and Rationale: The background provides a good context for the study, emphasising the importance of glycemic control in diabetic CKD patients on hemodialysis.

Literature Review: The introduction references relevant studies and statistics, but could benefit from a more detailed discussion of the specific challenges faced in Tanzania compared to other regions.

Methodology

Study Design and Setting: The study design and setting are well described. The choice of centers and the rationale for their selection are clearly explained.

Inclusion and Exclusion Criteria: The inclusion and exclusion criteria are appropriate. However, the exclusion criteria for patients without recent HbA1c results might have been further elaborated upon, considering potential biases.

Sample Size Calculation: The sample size calculation is justified based on previous prevalence data. It would be helpful to mention any adjustments or considerations made during the study due to recruitment challenges.

Data Collection: The data collection process is well detailed. The use of both primary and secondary data is a strength, but more information on the validation of the semi-structured questionnaire could add to the credibility.

Results

Presentation of Findings: The results are clearly presented with appropriate use of tables. Tables 1 and 2 effectively summarise baseline characteristics and clinical data.

Statistical Analysis: The statistical methods used are appropriate. The use of univariate and multivariable regression models is well justified, but providing more details on how potential confounders were controlled in the analysis would be beneficial.

Clarity in Results: The results section is clear, but a more explicit connection between the statistical findings and their clinical implications could enhance understanding.

Discussion

Interpretation of Findings: The discussion effectively interprets the results and places them in the context of existing literature. However, some parts are repetitive, particularly concerning the importance of glycemic control.

Strengths and Limitations: The discussion of the study's strengths and limitations is comprehensive. However, the potential impact of the exclusion criteria (patients without HbA1c results) on the findings could be more thoroughly examined.

Future Directions: Suggestions for future research are provided, which is good. Including more specific recommendations for clinical practice in Tanzania could add value.

References

Relevance and Recency: The references cited are relevant and recent, supporting the study's context and findings. Ensure that all references are formatted consistently according to the journal's guidelines.

Formatting and Language

Language and Grammar: The manuscript is generally well-written, but there are minor grammatical errors and typographical mistakes. Proofreading for these issues is recommended.

Formatting: Ensure consistency in formatting, particularly in tables and figure legends.

Conclusion

Summary of Key Findings: The conclusion summarizes the key findings well. It reiterates the high prevalence of poor glycemic control and the significant determinants identified.

Implications for Practice: The conclusion could better highlight the implications for clinical practice and policy in Tanzania.

Specific Comments

Table 1 and 2 Titles: Ensure that all tables have descriptive titles that indicate their content clearly.

Overall

The manuscript is a valuable contribution to understanding glycemic control in diabetic CKD patients on hemodialysis in Tanzania. With some minor revisions and additional clarifications, it will be well-suited for publication.

6. PLOS authors have the option to publish the peer review history of their article (what does this mean? ). If published, this will include your full peer review and any attached files.

**Do you want your identity to be public for this peer review?** For information about this choice, including consent withdrawal, please see our Privacy Policy .

Reviewer #1: No

Reviewer #2: **Yes: ** Ali Ahsan

---

## [Author Response · Author response to Decision Letter 1]

17 Sep 2024

All comments have been addressed and attached.

---

## [Editor Report · Decision Letter 1]

8 Oct 2024

PONE-D-24-23929R1Prevalence and determinants of poor glycemic control among diabetic chronic kidney disease patients on maintenance hemodialysis in TanzaniaPLOS ONE

Dear Dr. Mutagonda,

Thank you for submitting your manuscript to PLOS ONE. After careful consideration, we feel that it has merit but does not fully meet PLOS ONE’s publication criteria as it currently stands. Therefore, we invite you to submit a revised version of the manuscript that addresses the points raised during the review process.

We look forward to receiving your revised manuscript.

Kind regards,

Diana Laila Ramatillah, PhD

Academic Editor

PLOS ONE

**Journal Requirements:**

**Additional Editor Comments:**

Please revise the manuscript based on the comments

---

## [Editor Report · Decision Letter 2]

30 Oct 2024

PONE-D-24-23929R2Prevalence and determinants of poor glycemic control among diabetic chronic kidney disease patients on maintenance hemodialysis in TanzaniaPLOS ONE

Dear Dr. Mutagonda,

Thank you for submitting your manuscript to PLOS ONE. After careful consideration, we feel that it has merit but does not fully meet PLOS ONE’s publication criteria as it currently stands. Therefore, we invite you to submit a revised version of the manuscript that addresses the points raised during the review process.

We look forward to receiving your revised manuscript.

Kind regards,

Diana Laila Ramatillah, PhD

Academic Editor

PLOS ONE

Journal Requirements:

Additional Editor Comments:

We can accept after minor revision

---

## [Editor Report · Decision Letter 3]

20 Jan 2025

Prevalence and determinants of poor glycemic control among diabetic chronic kidney disease patients on maintenance hemodialysis in Tanzania

PONE-D-24-23929R3

Dear Dr. Mutagonda

We’re pleased to inform you that your manuscript has been judged scientifically suitable for publication and will be formally accepted for publication once it meets all outstanding technical requirements.

Kind regards,

Diana Laila Ramatillah, PhD

Academic Editor

PLOS ONE
---

## [Editor Report · Acceptance letter]

PONE-D-24-23929R3

PLOS ONE

Dear Dr. Mutagonda,

I'm pleased to inform you that your manuscript has been deemed suitable for publication in PLOS ONE. Congratulations! Your manuscript is now being handed over to our production team.

Kind regards,

on behalf of

Prof Diana Laila Ramatillah

Academic Editor

PLOS ONE